



# Multi-site learning for hydrological uncertainty prediction: the case of quantile random forests

Taha-Abderrahman El Ouahabi[1], François Bourgin[1], Charles Perrin[1], and Vazken Andréassian[1]

[1]Université Paris-Saclay, INRAE, HYCAR, Antony, France

**Correspondence:** Taha-Abderrahman El Ouahabi (taha-abderrahman.el-ouahabi@inrae.fr)

**Abstract.**

To improve hydrological uncertainty estimation, recent studies have explored machine learning (ML)-based post-processing approaches that enable both enhanced predictive performance and hydrologically informed probabilistic streamflow predictions. Among these, random forests (RF) and their probabilistic extension, quantile random forests (QRF), are increasingly

used for their balance between interpretability and performance. However, the application of QRF in regional post-processing settings remains unexplored. In this study, we develop a hydrologically informed QRF post-processor trained in a multi-site setting and compare its performance against a locally (at-site) trained QRF using probabilistic evaluation metrics. The QRF framework leverages simulations and state variables from the GR6J hydrological model, along with readily available catchment descriptors, to predict daily streamflow uncertainty. Our results show that the regional QRF approach is beneficial for

hydrological uncertainty estimation, particularly in catchments where local information is insufficient. The findings highlight that multi-site learning enables effective information transfer across hydrologically similar catchments and is especially advantageous for high-flow events. However, the selection of appropriate catchment descriptors is critical to achieving these benefits.

## 1  Introduction

### 1.1  On the need for quality uncertainty estimates

Providing quality uncertainty estimates for streamflow predictions is critically important, particularly in applications such as operational drought simulation, water resource management, and flood mitigation where significant stakes are involved (Hwang et al., 2019; White et al., 2017). Poorly quantified or overly confident predictions can lead to misinformed decisions, potentially resulting in economic losses, infrastructure damage, or even threats to public safety. To address this, various approaches have

been proposed in the hydrological community for streamflow uncertainty quantification, including multi-model ensembles (Georgakakos et al., 2004; Troin et al., 2021), Bayesian inference (Kuczera and Parent, 1998; Bates and Campbell, 2001), and hydrological error modeling (Krzysztofowicz, 1999; Todini, 2008; Solomatine and Shrestha, 2009; Bennett et al., 2021), referred to as post-processing.





Hydrological uncertainty post-processing techniques aim to statistically model hydrological predictive distributions and
were adopted early through methods such as the hydrological uncertainty processor (HUP) (Krzysztofowicz, 1999) and model
conditional processor (MCP) (Todini, 2008), but recent machine learning (ML)-based approaches have emerged as powerful
tools for hydrological post-processing. Although less interpretable, ML-based approaches can potentially produce reliable and
improved informative uncertainty estimates (Papacharalampous and Langousis, 2022; Tyralis and Papacharalampous, 2024).
Methods such as quantile regression (QR) (Tyralis et al., 2019; Papacharalampous and Langousis, 2022), conformal prediction
(Auer et al., 2024), and random forests (Zhang et al., 2023) have been used for streamflow post-processing with promising
results. However, ML algorithms can produce different uncertainty estimates depending on how they are trained — particularly
on which catchments are included in the training dataset. Since hydrological conditions vary significantly across catchments,
the selection of catchments used for training can influence the uncertainty estimates of the ML model. In our study, we aim to
explore whether including different catchments (multi-site learning) may improve ML-based post-processing for uncertainty
estimation, and specifically for the quantile random forests (QRF) model.

## 1.2 Machine learning-based post-processors

Random forest (RF) (Breiman, 2001) and its probabilistic variant, quantile random forest (QRF) (Meinshausen and Ridgeway,
2006) are extensively used and are considered state of the art in many hydrological applications. Recently, Zhang et al. (2023)
compared the QRF model and the countable mixtures of asymmetric Laplacians long short-term memory (CMAL-LSTM)
model to probabilistically post-process streamflow simulations across 522 catchments. The QRF and CMAL-LSTM models
were comparable in terms of uncertainty estimates, but the CMAL-LSTM deep learning (DL) model performed better in catch-
ments with large flow accumulation areas. QRF has also been applied in hydrologically informed post-processing approaches.
Shen et al. (2022) use an RF framework and leverage internal state variables to correct PCR-Global (PCRaster Global Water
Balance, a global hydrological model) simulations at three stations in the Rhine Basin. They found that the use of hydrologi-
cal model states as input features of RFs provides additional information that may not be included in the model simulations.
However, challenges remain, particularly in modeling errors during high streamflow periods. Magni et al. (2023) expand the
same approach at the global scale, using PCR-Global model simulations and internal states, in conjunction with static catch-
ment attributes, to train a single RF model on a global database of streamflow simulations and measurements. They found that
improvements were independent of the availability of streamflow data, indicating the power of regional learning methods in
poorly gauged and ungauged catchments.

Prediction in ungauged basins is not the only benefit of training a single ML model on data from multiple catchments.
Kratzert et al. (2024) advocate the use of regional approaches to fit a deterministic Long Short-term Memory (LSTM) DL model
for streamflow simulations. They found that larger LSTM models trained on all available basins outperform smaller models
trained on a limited set of catchments. This is because, for some ML approaches, models calibrated on larger training datasets
can outperform smaller and more specialized models (Montero-Manso and Hyndman, 2021). Furthermore, Johnson et al.
(2023) found that hydrological model performance depends on basin attributes, indicating the presence of regional and spatial



bias. This can be harnessed to improve uncertainty estimation using a post-processing model with spatial parametrization and trained on hydrologically diverse rainfall-runoff responses.

### 1.3 Scope of this study

This study focuses on the added value of a regional post-processing approach for hydrologically informed quantile random forests (QRF). The main contributions of this work are: (i) to understand the impact of including different catchments in the training process of QRF (multi-site) and to test if it can improve its uncertainty estimates and (ii) to investigate the importance of spatial catchment descriptors for these multi-site QRFs.

For this, we use temporally varying information (predicted streamflows and model states) and spatially varying catchment
characteristics. We chose to focus on QRF due to its balance of performance, interpretability, and popularity in the hydrological community. To the best of our knowledge, no prior study has explored the impact of multi-site learning with the QRF algorithm for uncertainty estimation, particularly when post-processing a hydrological model calibrated separately for each catchment. To that end, we fit different QRF variants on the internal states of a hydrological model, on meteorological variables, and on readily available catchment characteristics. The proposed regional QRF variants are evaluated across a large sample of 564
French catchments to identify when multi-site learning may be beneficial and to offer practical considerations for multi-site QRF application.

The paper is organized as follows: We first introduce the dataset and describe the QRF algorithm, its variants, and the probabilistic evaluation framework. Then, we present and discuss the results before summarizing the key findings along with implications for future work.

## 2 Dataset

### 2.1 A dataset of 564 French catchments

We used a set of 564 catchments distributed throughout France (Fig. 1). These catchments represent a wide range of hydrological regimes and simulation contexts. We selected these catchments from the CAMELS-FR hydroclimatic dataset (Delaigue et al., 2024). The criteria for selecting these catchments were as follows: (i) low anthropogenic influence, (ii) good data quality
for all flow regimes, and (iii) an available time series longer than 21 years. Streamflow data were obtained from the national HydroPortail archive (Leleu et al., 2014; Dufeu et al., 2022) at a daily time step for the period 1977–2021. Meteorological forcings (precipitation and temperature) were provided by Météo-France's daily SAFRAN grid reanalysis (Vidal et al., 2010). Potential evaporation (PET) is calculated using the formula proposed by Oudin et al. (2005). Since our interest is in developing a multi-site QRF post-processor, we used several static basin-averaged attributes describing climate, topography and geology.
All of these attributes are included in the CAMELS-FR dataset.



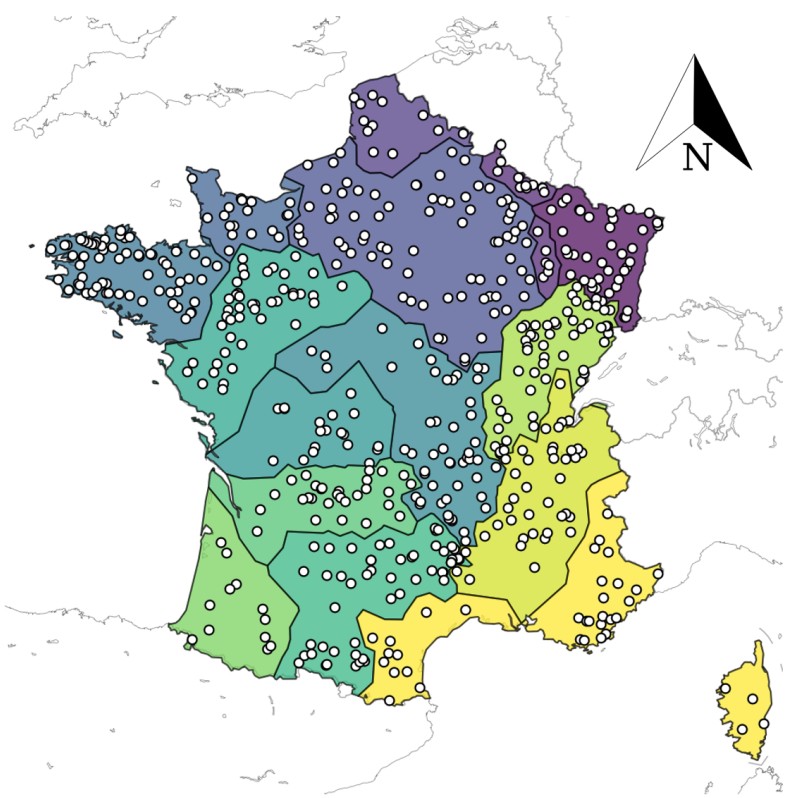

**Figure 1.** Location of the 564 catchment outlets. Plotted regions represent the hydroclimatological catchment groups used in the study.

To achieve separation between hydrological and statistical calibration, data time series were split into two periods: the first period (1977–1989), during which the hydrological model was calibrated, and the second period (1990–2021), during which QRF variants were trained and tested. The second period followed a classic train–validation–test splitting procedure:

– **P1**: training period from 1990 to 2004, used to train the QRF post-processor.

– **P2**: validation period from 2005 to 2009, used to select the hyperparameters of the QRF post-processor.

– **P3**: testing period from 2010 to 2021, used to test the performance of the QRF variants on new data.

Finally, to ensure that sufficient data are available during the training period, all catchments in the dataset have at least 7 years of discharge data available during that period.





## 2.2 Methods

**2.3 Hydrological model**

We used discharge simulations obtained with the GR6J rainfall–runoff model (Pushpalatha et al., 2011), a daily 6-parameter conceptual lumped model. GR6J has been applied in several studies across a large number of catchments and hydroclimatic contexts (e.g. Poncelet et al., 2017; Golian et al., 2021; Tanguy et al., 2023). The GR6J model is based on several state variables that control its simulations, in particular the production and routing store levels, and intercatchment exchange fluxes. We intend

to use these state variables as predictors in the QRF algorithm. Shen et al. (2022) successfully used internal state variables as predictors in an RF framework to correct hydrological model errors. They found that internal state variables provided valuable information for the RF, enabling it to detect and correct for systematic hydrological model errors. To account for the influence of snow of some catchments, we incorporated Cemaneige (Valéry et al., 2014), a snow accumulation and melt model, with constant parameters for all catchments.

The Cemaneige-GR6J model was calibrated using the airGR R package (Coron et al., 2017, 2023) with the built-in calibration algorithm. To ensure good performance across a wide range of streamflow conditions, the target optimization criterion was a combination of KGE criteria (Gupta et al., 2009; Kling et al., 2012): an equal weighting of KGE criteria with a power of 0.5 and -0.5 prior transformations on streamflow.

### 2.4 Feature selection and data transformations

**2.4.1 Target variable**

For the purposes of this study, we model the probabilistic distribution of hydrological model errors. Since these errors are skewed and non-Gaussian (e.g. Evin et al., 2014), we applied a logarithmic transformation to improve the training process:

$$\epsilon_t = log(\frac{Q_t^{obs} + \delta}{Q_t^{sim} + \delta}) \tag{1}$$

where $\epsilon_t$ is the target variable of our study and represents the prediction error, $Q_t^{obs}$ and $Q_t^{sim}$ indicate observed and simulated

streamflows, respectively (mm/day), and $t$ represents the time index with a temporal step of 1 day. $\delta$ is an offset parameter to avoid zero streamflow values and is unique for each catchment. It was calculated following recommendation in Pushpalatha et al. (2012). The use of $\delta$ is especially relevant in this study due to the application of the logarithmic transformation.

The input predictors (or features) in the QRF models are listed in Table 1. These features can be broadly categorized into two groups of approximately equal size: (i) time series data (dynamic features) that capture temporal variability, and (ii) catchment

descriptors (static features) that enable spatial identification of catchments.





### 2.4.2 Dynamic features

The proposed QRF framework post-processes GR6J simulations and uses hydrological model outputs and state variables along with meteorological inputs (precipitation and temperature). Streamflow uncertainties are known to be autocorrelated (Evin et al., 2014) with strong autoregressive (AR) and memory effects. Consequently, lagged observed streamflow (Zhang et al.,

2023; Pham et al., 2020) is a popular input feature for RF-based post-processing. In the simulation context of this study, streamflow is not available and we use state features from the GR6J model to provide additional information to QRF. Although some of the features in Table 1, such as simulated flows and production store, are strongly autocorrelated, we assume that the additional information still leads to improved uncertainty estimates compared to using model simulations alone. Similarly to (Shen et al., 2022) we include other temporal information in QRF through transformed features: (i) increment features of

simulated streamflow, production, and routing store levels to help capture the dynamics of the hydrograph (rising and falling limbs etc.) and (ii) moving averages of meteorological features to highlight general trends. This feature engineering step can be relevant for RF-based algorithms in a time series context, because QRF does not create temporal memory or embeddings as is the case for AR models and LSTM neural networks (Evin et al., 2014; Li et al., 2016; Kratzert et al., 2018).

### 2.4.3 Static features

To take into account spatial heterogeneity, catchment descriptors are used for the multi-site QRF variants. The static features include: (i) average catchment attributes such as catchment area and aridity index. We chose to keep thirteen relevant catchment attributes following the recommendations of Jehn et al. (2020); (ii) scale features of errors, simulated flows, and observed streamflows. These scale features provide additional unique indicators of catchment characteristics. Montero-Manso and Hyndman (2021) found that combining individual time series features such as catchment attributes with scale features can improve

the performance of ML models in a deterministic setting. Similar improvements are expected for QRF in the setting of hydrological uncertainty estimation. But, it is important to note that these scale features are not available for prediction in the context of ungauged catchments, as they are calculated based on observed streamflows.

### 2.5 QRF: how to fit the algorithm?

Random forest Breiman (2001) is a non-parametric ensemble tree-based model that offers good performance and provides

certain interpretability through its feature importance estimates (Breiman et al., 1984; Breiman, 2001). RF and its probabilistic version QRF are used extensively in the hydrometeorological domain. An important advantage of QRF is that it provides full distributional estimates without the need to estimate each quantile separately, as is required in quantile regression (Tyralis et al., 2019; Papacharalampous and Langousis, 2022). QRF has been applied to complex and heteroscedastic cases, including hydrometeorological ensemble forecasts (Taillardat et al., 2016; Tiberi-Wadier et al., 2021; Teja et al., 2023), post-processing

of streamflow simulation (Zhang et al., 2023), and estimation of the limits of acceptability for hydrological models (Gupta et al., 2024). Further details on the construction of RF and QRF can be found in (Louppe, 2014; Meinshausen and Ridgeway, 2006), but QRF can be viewed as an analog method (Delle Monache et al., 2013; Hu et al., 2023) that performs a weighted



**Table 1.** Features used in the study

| Features | Unit | Description | Type |
|---|---|---|---|
| PotEvap | mm/day | potential evapotranspiration | |
| Precip | mm/day | precipitation | |
| AE | mm/day | actual evapotranspiration | |
| Prod | mm | production store | |
| Rout | mm | routing store | |
| AExch | mm/day | intercatchment exchange | |
| Qsim | mm/day | simulated flows | |
| Delta_sim7 | mm/day | 7-day difference in simulated flows | |
| Delta_sim1 | mm/day | 1-day difference in simulated flows | |
| Delta_rout7 | mm | 7-day difference in routing store | **Dynamic features** |
| Delta_rout1 | mm | 1-day difference in routing store | |
| Delta_prod1 | mm | 1-day difference in production store | |
| Prec_sold_frac | - | fraction of solid precipitation | |
| Temp | °C | temperature | |
| SWI_ISBA | - | soil wetness index | |
| Rolling_temp | °C | moving average of temperature | |
| Rolling_precip | mm/day | moving average of precipitation | |
| Rolling_sold_frac | mm/day | moving average of solid precipitation | |
| Month_of_year | - | annual cycle (cosine term) | |
| top_drainage_density | - | drainage density | |
| sit_area_topo | km$^2$ | topographic catchment area | |
| hyd_bfi_pelletier_pet_ou | - | baseflow index | |
| cli_prec_mean | mm/day | mean daily precipitation | |
| cli_pet_ou_mean | mm/day | mean daily potential evapotranspiration | |
| cli_temp_mean | °C | mean daily temperature | |
| cli_aridity_ou | - | aridity index | |
| cli_psol_frac_safran | - | mean fraction of solid precipitation | |
| cli_prec_freq_high | - | frequency of high-precipitation days | |
| cli_prec_freq_low | - | frequency of dry days | |
| top_altitude_mean | m | mean catchment elevation | **Static features** |
| cli_prec_season_pet_ou | - | seasonality index | |
| Response_Time | days | Response time based on the X4 parameter from GR6J | |
| mean_Qsim | mm/day | mean Qsim | |
| std_Qsim | mm/day | standard deviation Qsim | |
| mean_Qobs | mm/day | mean Qobs | |
| std_Qobs | mm/day | standard deviation Qobs | |
| mean_Error_log | - | mean error log | |
| std_Error_log | - | standard deviation error log | |
| Region indicator | - | Hydroclimatological region of the catchment | |





nearest-neighbor search for analogous events. Similarly to a classic RF, QRF grows a number of trees $n$, with each tree trained on a bootstrapped subsample of the original training data.

Individual trees are trained according to the Breiman et al. (2017) algorithm by minimizing a loss function and making successive splits with a predefined number of features $p$. This tree-building process enables QRF to account for strongly correlated features, which is important given the strong correlation of some of the features used. For the purpose of this study, we use mean squared error (MSE) as loss function to calculate the homogeneity of each group. The procedure continues recursively, with each resulting group split further until a minimum number of data points $m$ in child splits is reached.

In the classic Random Forest (RF) algorithm, predictions from individual trees are averaged to produce a single deterministic output. In contrast, Quantile Regression Forest (QRF) leverages the leaf nodes of trees to compute proximity measures between a test input and training instances. For a prediction at time $t$ and given input $x_t$, each QRF tree is traversed using binary splits to reach a corresponding leaf node. A proximity weight $\omega_i(x_t)$ is then defined for each training instance $i$ (Meinshausen and Ridgeway, 2006), which is then used to estimate the cumulative distribution function (CDF) of the prediction uncertainty:

$$\hat{F}(\epsilon|x_t) = \sum_{i=1}^{n} \omega_i(x_t)\mathbb{1}_{\{\epsilon_i \leq \epsilon\}} \tag{2}$$

where $\omega_i(x_t) > 0$ and $\sum_{i=1}^{n} \omega_i(x_t) = 1$, $\epsilon_i$ denotes the hydrological error of training instance $i$, and $\hat{F}$ is the estimated CDF of uncertainty for $x_t$.

To provide reliable and sharp uncertainty estimates, we considered three hyperparameters for optimization: (i) The number of trees $n$, which controls precision and stability. A larger number of trees improves the quality of uncertainty estimates, but

improvements diminish as computational cost increases, especially in larger models. (ii) The minimum number of samples at child nodes $m$, which affects tree depth and strongly impacts reliability and sharpness. Setting high values for the minimum samples per leaf might yield high reliability, but can lead to poor performance, as the trees are too general and information is lost. Low values result in overfitting and yield unreliable uncertainty estimates. (iii) The number of features per split, $p$, which also shapes the QRF uncertainty estimates. Higher values can lead to under-dispersed uncertainties, while lower values

may reduce sharpness. Further details on hyperparameter values and selection are provided in Appendix A. Additionally, we also use a K-nearest neighbor (K-NN) (Wani et al., 2017) approach as a benchmark for the QRF methods used in the study. Like QRF, K-NN aims to find analogous events but based on the Euclidean distance between features. Here, K-NN is fitted locally on the same variables as for QRF. Further details on the fitting process and the hyperparameters used are provided in Appendix B.

Given equation (2), the estimated CDF is bounded by the learning sample. QRF is unable to predict a quantile higher than the maximum observed in the training sample, which implies that QRF trained on a single basin is constrained by the range of errors in its training data. A more hydrologically diverse training dataset would alleviate this problem and enable QRF to adapt to more extreme events, provided that QRF is able to use the additional information properly.





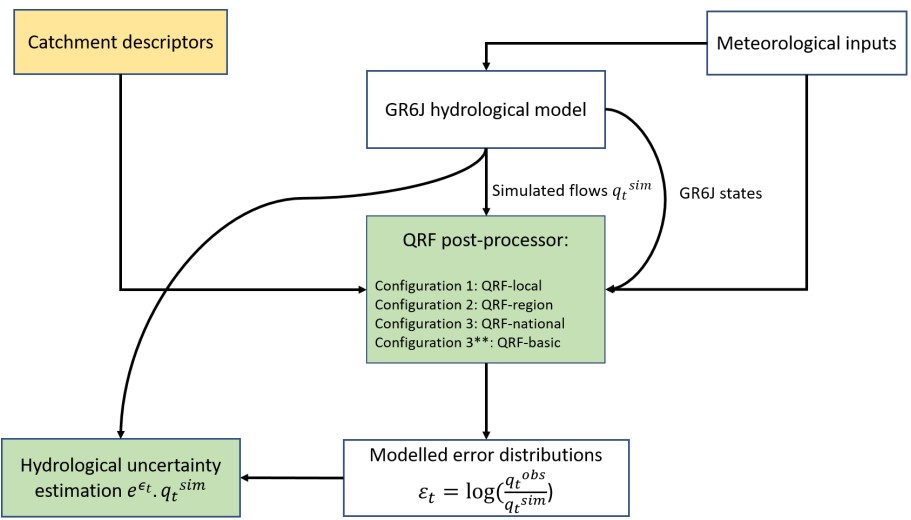

**Figure 2.** Schematic overview of implementing QRF based post-processors. Configurations 1, 2, and 3 represent the regional information used in QRF post-processing. A description of the features used is provided in Table 2.

## 2.6 QRF variants

Figure 2 presents the framework employed for the QRF configurations analyzed in this study. The local approach (QRF-local) refers to training the QRF algorithm on data from a single basin. Given their construction, spatial features are constant for each individual catchment and only time-series features can be fed to QRF in the local setup. QRF-local yields 564 independently trained QRF models, each specific to its respective catchment. Next, spatial variability is added as we extend QRF to a multi-site setting. The objective here is twofold: (i) to examine whether spatial diversity can improve the uncertainty estimates of

QRF. This can be challenging, particularly since the used GR6J hydrological model is calibrated on an on-site basis; and (ii) to determine the optimal number of catchments to include in the training set for effectively capturing hydrological diversity and improving QRF predictions. We test QRF with two spatial settings: (i) a regional approach (QRF-region), where QRF is trained on data from catchments that are geographically close and thus potentially have similar error dynamics. In total, 15 regional QRF models are developed for the hydrological regions of the study, based on hydroclimatological groupings of French

catchments; (ii) a global approach (QRF-national), in which a QRF is trained on data from all catchments in the dataset. Both static and dynamic descriptors are used in the training process of QRF-region and QRF-national. However, QRF-national uses the catchment's hydrological region as an additional input feature, which cannot be used for QRF-region. Intuitively, in cases where QRF is unable to transfer information from different basins or when there are no useful analogs in similar catchments, QRF-local would yield better performance, as no information from other catchments is used to build the model. To assess

the usefulness of static features for the multi-station QRF setup, we included QRF-basic, a global QRF approach fitted on all catchments of the study, but only with dynamic features. This experiment is expected to highlight whether dynamic time series



features are sufficient to improve multi-site QRF predictions or that static features are essential for multi-site post-processing. Table 2 presents the features used in the three configurations.

It is worth mentioning that in multi-site setups, the standardization procedure is an essential step that enables QRF to determine analogs across a set of diversified catchments, as the scales of streamflows and dynamic features (GR6J states and transformed variables) vary significantly. Standardization is important for a meaningful training process and for the identification of adequate analogous events. Initially, we standardized input data via the popular standard scaling method (Hastie et al., 2001), which transforms dynamic features – for each catchment – so that the average and standard deviation are set to 0 and 1, respectively. However, the method resulted in inconsistencies for catchments with outliers, as the standard deviation is sensitive to extreme values. To solve this issue, we opted for robust standardization (Hastie et al., 2001), which removes median values of dynamic features and the target errors $\epsilon_t$ defined in Section 2.3.

**Table 2.** QRF variants of the study

| Configurations | Dynamic features | Static features | Hydroclimatological group | Number of models |
|---|---|---|---|---|
| QRF-local | ✓ | | | 564 |
| QRF-regional | ✓ | ✓ | | 15 |
| QRF-national | ✓ | ✓ | ✓ | 1 |
| QRF-basic | ✓ | | | 1 |

# 3 Assessment criteria

In this section, we present the probabilistic metrics used to evaluate the three variants of QRF. The followed criterion for probabilistic predictions conforms to Gneiting et al. (2007) objective of maximizing calibration (reliability) subject to sharpness. Reliability refers to the statistical consistency between probabilistic predictions and observed streamflow values, while sharpness is a property of predictions exclusively and refers to the dispersion or tightness of the predicted uncertainty distributions. We used distributional, interval-based, and deterministic evaluation criteria to obtain a holistic point of view of the proposed QRF variants. All QRF variants in this study predict, at each time step, 200 quantile members equally spaced between 0.005 and 0.995. All of the scores used are calculated using the EvalHyd (Hallouin et al., 2023) python library.

## 3.1 Distributional metrics: alpha score, sharpness, and CRPSS

The alpha score (Renard et al., 2010) targets reliability. It calculates the closeness of predicted uncertainty distributions to the statistical distribution of observed streamflows. The values of the metric range from 0 (worst reliability) to 1 (perfect reliability). For sharpness, we used a skill score to measure sharpness, defined as the ratio between the sharpness metric and the reference distribution. In our case, the reference distribution is the empirical distribution of observed streamflows during the training period. The sharpness metric is the continuous ranked probability score (CRPS) (Gneiting et al., 2005a) of the





estimated predictions compared to its median (Appendix C). A perfect point forecast is assigned a score of 1; positive values indicate better performance compared to the climatological distribution, and negative values indicate worse performance. We also use the CRPS, which is a popular scoring measure for assessing reliability and sharpness simultaneously (Gneiting et al., 2005b). For given probabilistic prediction members, the CRPS calculates the difference between the cumulative distribution function (CDF) of the uncertainty estimates and the observed streamflow values. We use the CRPS skill score relative to the climatological distribution to obtain a positively oriented score (a higher score is better). The CRPSS value for perfect point prediction is 1; positive values indicate better performance than the climatological distribution, and negative values indicate worse performance, as per Appendix C.

## 3.2 Coverage ratio, average interval width, and Winkler score

To provide a more comprehensive assessment of predictive uncertainty, evaluation metrics were calculated for prediction intervals at the 90% and 95% levels. The coverage ratio (CR) is a measure of reliability that counts the number of observations that lie within the prediction intervals. Values closest to the desired coverage level (i.e., 90% or 95%) are more reliable. To assess the sharpness, we employ the average width metric (AW), which corresponds to the average width of the prediction interval during the evaluation period. We also evaluate the Winkler score (WS), which simultaneously includes both criteria and enables an easy comparison between the variants of the study. Both AW and WS are presented as skill scores - AW skill score (AWSS) and Winkler skill score (WSS) - relative to the climatological distribution.

## 3.3 Deterministic metrics

Although the main focus of this study is probabilistic post-processing, some decision-makers may require deterministic predictions. Therefore, we also evaluate mean predictions to compare the different post-processing variants of the study. We use the popular Nash–Sutcliffe efficiency (NSE) (Nash and Sutcliffe, 1970) and Kling–Gupta efficiency (KGE) (Gupta et al., 2009) metrics to gauge the quality of deterministic predictions in multi-site learning setups.

## 4 Results

In this section, we compare each QRF variant according to its performance during the testing period. We investigate flow ranges in which multi-site learning is preferable, and we explore the importance of including catchment descriptors for regional QRFs. The results for the K-NN approach can be found in Appendix B. Figure. 3 illustrates the uncertainty estimates of QRF-local for a randomly selected catchment.

## 4.1 Reliability, sharpness, and CRPSS

We first present our results with distributional metrics in Figure 4, which shows the cumulative distribution of reliability, sharpness, and CRPSS for the 564 catchments of the study. QRF-region and QRF-national slightly improve reliability compared to QRF-local. However, multi-site learning does not yield better alpha scores for well-calibrated stations with QRF-local. Fig-



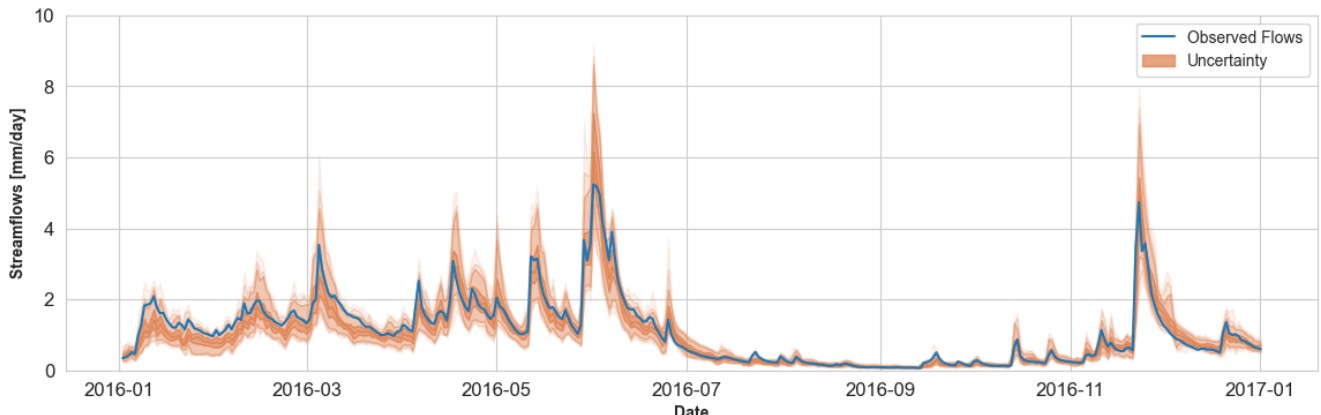

**Figure 3.** Example randomly selected for (station code K287191001 at Giroux with 756 km$^2$) a catchment between $1^{st}$ January 2016 and $1^{st}$ January 2017 comprising both high- and low-flow events. Uncertainty estimates from QRF-local are plotted with observed flows. Darker orange shades indicate regions of higher probability (25% and 75% quantiles). Lighter regions indicate low probability quantiles (5% and 95% quantiles).

ure 5 shows a direct comparison of the proposed variants and indicates that improvements were most noticeable for catchments where QRF-local provided low reliability, i.e., 25% of the basins with the lowest alpha scores (the 25% quantiles of the alpha score were 0.742 for QRF-local and 0.76 and 0.76 for QRF-region and QRF-national, respectively). In terms of sharpness, the different QRF variants performed similarly, which is interesting given that multi-site setups significantly improve CRPSS

values. Among the QRF variants, QRF-national generally outperformed QRF-local, improving CRPSS by approximately 2%, except in the case of four catchments, where QRF-local performed significantly better. Additionally, QRF-region improved CRPSS for 69% of the catchments compared to QRF-local, while QRF-national showed improvements in 88% of the catchments. Overall, the improvements are less apparent for reliability, but multi-site QRFs seems to improve performance for catchments with initially poor calibration in the local setup. Given that the sharpness metric was nearly identical across the

QRF variants in the study, we suspect that the CRPSS improvements are mainly due to improvements in reliability.

## 4.2  Interval metrics

We now consider the 90% interval metrics. Figure 6 represents the box plots across the 564 catchments of the study for coverage ratio, average interval width, and Winkler skill scores. The multi-site learning setup was beneficial for QRF and enabled better coverage. For instance, the median coverage ratios were set at 0.87, 0.89, and 0.89 for QRF-local, QRF-region, and QRF-

national, respectively. The improvements are also observed for Winkler skill score, where QRF-national provided the best results. However, the average interval width was similar for all the variants in the study, further indicating that improvements in multi-site learning in the case of QRF mainly relate to reliability. For the sake of completeness, we include interval metrics for the 95% predictive uncertainty interval in Appendix D, as the conclusions remain the same.





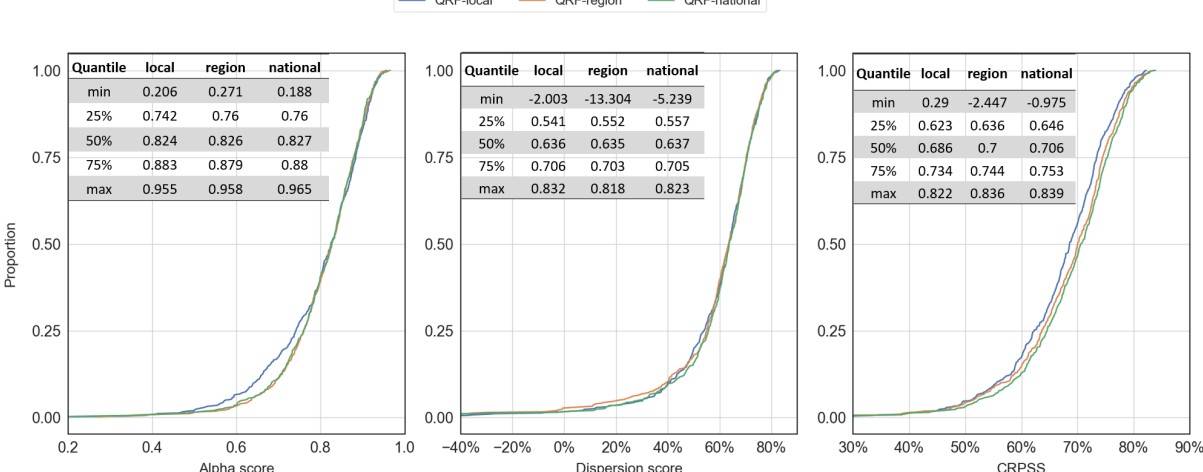

**Figure 4.** CDF of distributional metrics across the 564 catchments for the QRF variants in the study. The blue line represents the performance of QRF-local, orange represents QRF-region, and green represents QRF-national.

### 4.3 Deterministic metrics

Figure. 7 shows the cumulative distribution function for the deterministic metrics of Nash–Sutcliffe efficiency (NSE) and Kling–Gupta efficiency (KGE) scores. Multi-site learning improves NSE for most catchments, but for KGE, improvements are most apparent when the local approach yields low KGE values. For example, when investigating catchments at the lower 25% percentile, QRF-region and QRF-national improved the median KGE by 6%. However, for catchments where QRF-local provided decent KGE scores (top 25% performers), multi-site setups yielded similar scores to a single-basin approach.

This would highlight the equalizing effects of multi-site learning for QRF, as it is most impactful for catchments with poor single-basin post-processing. We argue that these results show: (i) the ability of QRF in its multi-site setup to identify and transfer useful information from neighboring catchments; (ii) although the improvements relate to both deterministic and uncertainty predictions, they are most significant for coverage ratio, CRPSS, WSS, NSE, and KGE. Building on these findings, we investigated whether these benefits were more pronounced under specific hydrological conditions.

### 285 4.4 How do QRF uncertainty estimates perform for different flow ranges?

Here, we aim to understand how the proposed QRF approaches perform across different flow ranges. Table 3 summarizes the average values of the alpha, dispersion, CRPSS, and interval scores for three flow groups: high ($> 67\%Q_{sim}$), medium ($> 34\%Q_{sim}$ and $< 66\%Q_{sim}$), and low flows ($< 33\%Q_{sim}$). Under low-flow conditions, the scores are similar, especially the sharpness and alpha scores. But for higher simulated flows, multi-site QRFs provide better calibration (alpha score and

coverage ratio) and better overall performance (CRPSS and WSS). Although QRF-local was able to provide narrower interval widths, especially for higher flows, it had lower reliability compared to multi-site QRFs. QRF-region and QRF-national adapt





**Figure 5.** Comparative plots between QRF variants in the study during the testing period. The first row shows the alpha score, the second row shows dispersion, and the third row shows CRPSS. The first column compares metric values for QRF-local vs. QRF-region, the second column for QRF-national vs. QRF-region, and the third column for QRF-national vs. QRF-region

to higher-flow ranges by providing wider uncertainty estimates and enable better calibration and conditionality, as reflected in the improved CRPSS and Winkler scores.




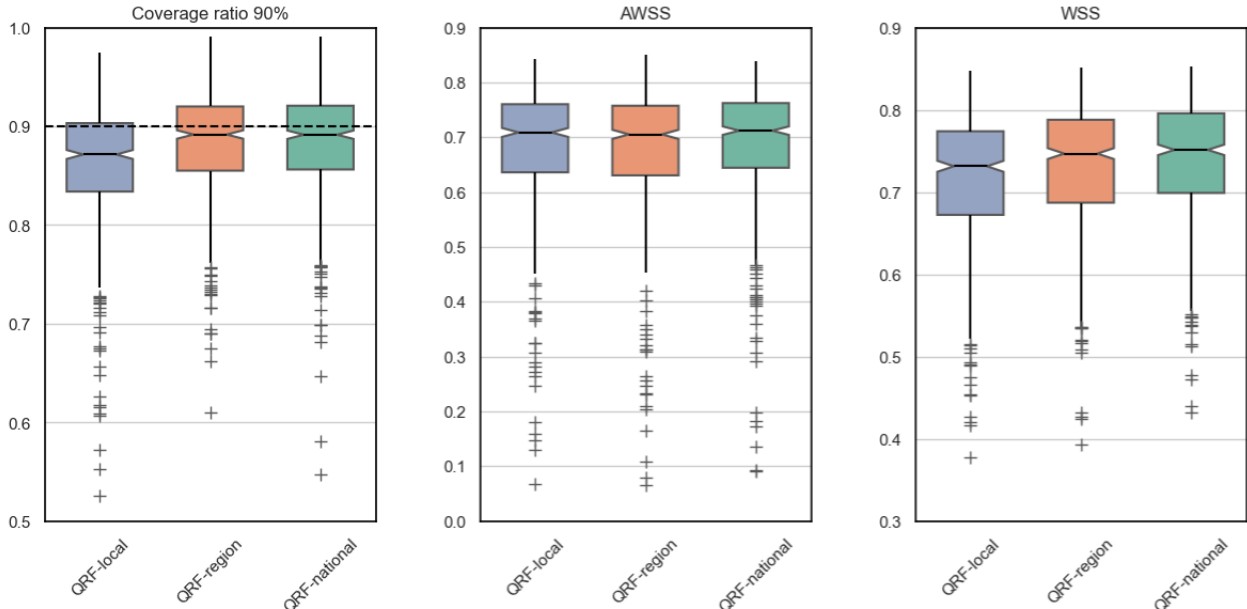

**Figure 6.** Box plots of 90% interval-based metrics across the 564 catchments of the study. Blue indicates the performance of QRF-local, orange indicates QRF-region, and green indicates QRF-national.

**Table 3.** Summary of average metrics for different QRF methods across the 564 catchments of the study. Three flow ranges are included: high, medium, and low simulated flows. The bold numbers indicate better performance in each group.

| Regime | QRF variant | Alpha score | Dispersion score | CRPSS | CR90.0 | AWSS | WSS |
|---|---|---|---|---|---|---|---|
| Low flows ($> 67\% Q_{sim}$) | QRF_local | **0.769** | **0.905** | 0.914 | 0.848 | **0.921** | 0.919 |
| | QRF_region | 0.767 | 0.901 | 0.919 | **0.874** | 0.918 | 0.925 |
| | QRF_national | 0.765 | **0.905** | **0.921** | 0.871 | 0.920 | **0.927** |
| Medium flows ($> 34\% Q_{sim}$ and $< 66\% Q_{sim}$) | QRF_local | 0.819 | **0.764** | 0.792 | 0.862 | **0.808** | 0.812 |
| | QRF_region | **0.833** | 0.752 | 0.797 | **0.882** | 0.798 | 0.821 |
| | QRF_national | 0.831 | 0.758 | **0.802** | **0.882** | 0.802 | **0.825** |
| High flows ($< 33\% Q_{sim}$) | QRF_local | 0.809 | -0.216 | 0.225 | 0.870 | **0.269** | 0.381 |
| | QRF_region | **0.827** | -0.342 | 0.247 | 0.889 | 0.154 | 0.391 |
| | QRF_national | 0.826 | **-0.213** | **0.264** | **0.890** | 0.249 | **0.427** |



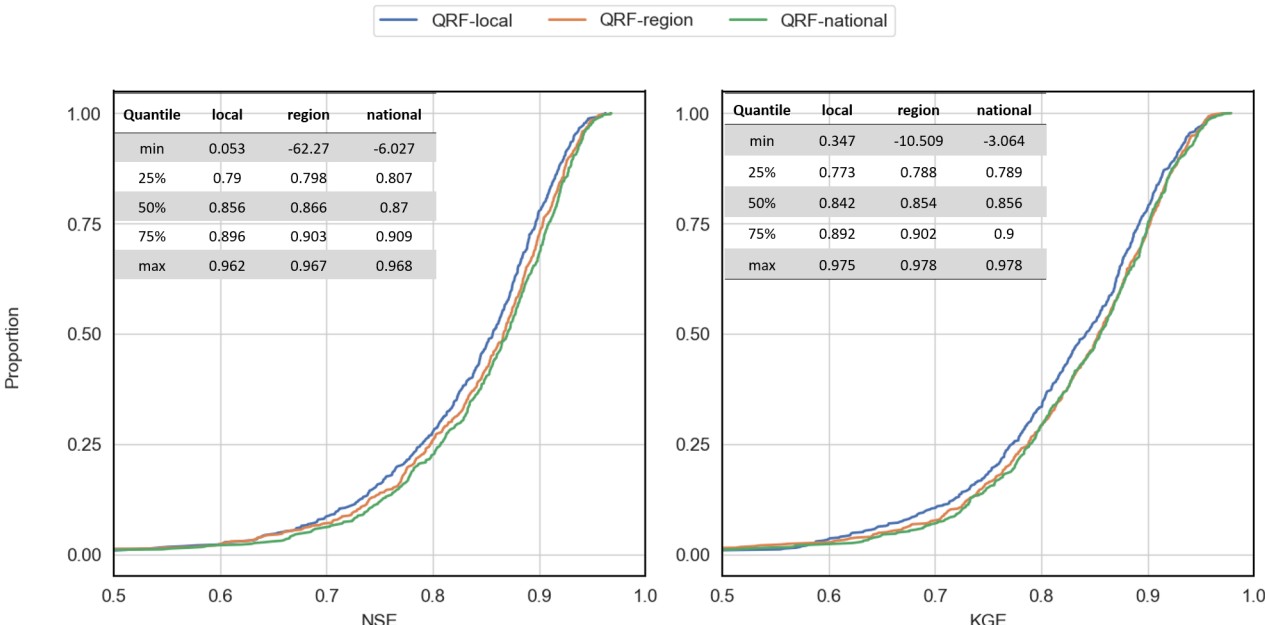

**Figure 7.** CDF of deterministic metrics across the 564 catchments for the QRF variants in the study. The blue line represents the performance of QRF-local, orange represents QRF-region, and green represents QRF-national.

## 4.5 Impact of static descriptors

To understand the impact of static catchment descriptors, Figure 8 illustrates the distributional metrics for QRF-national and QRF-basic. QRF-basic is a multi-site QRF trained across all catchments of the study and using the same features as for QRF-local (no static features). Notable differences between the two variants are observed: in terms of reliability (median 0.827 vs. 0.806 across all catchments), sharpness (0.637 vs. 0.614), and CRPSS (0.706 vs. 0.691). Largest difference are observed for CRPSS, as QRF-national was better for $80\%$ of the stations of the study. Furthermore, Figure D2 in Appendix D shows that QRF-basic had very similar CRPSS values as for QRF-local. These results suggest that the performance improvements in multi-site QRF models are not solely due to the inclusion of hydrological diversity in the training data. Static catchment descriptors play a significant role, and the selection of informative static features appears to be critical for effective multi-site QRF implementations.

## 4.6 Sensitivity to scale (potential for improving the performance of QRF-national)

Following the results of the previous section, we found that multi-site learning can significantly degrade the performance for four cases across distinct hydrological regions; an example of such catchments is presented in Figure. 9. To investigate this, Table 4 presents the differences in performance (alpha score and CRPSS) between QRF-national and QRF-local based on the variability of errors $\epsilon_t$ for three groups: group 1 is characterized by important error variability ($>1.5$), group 2 also



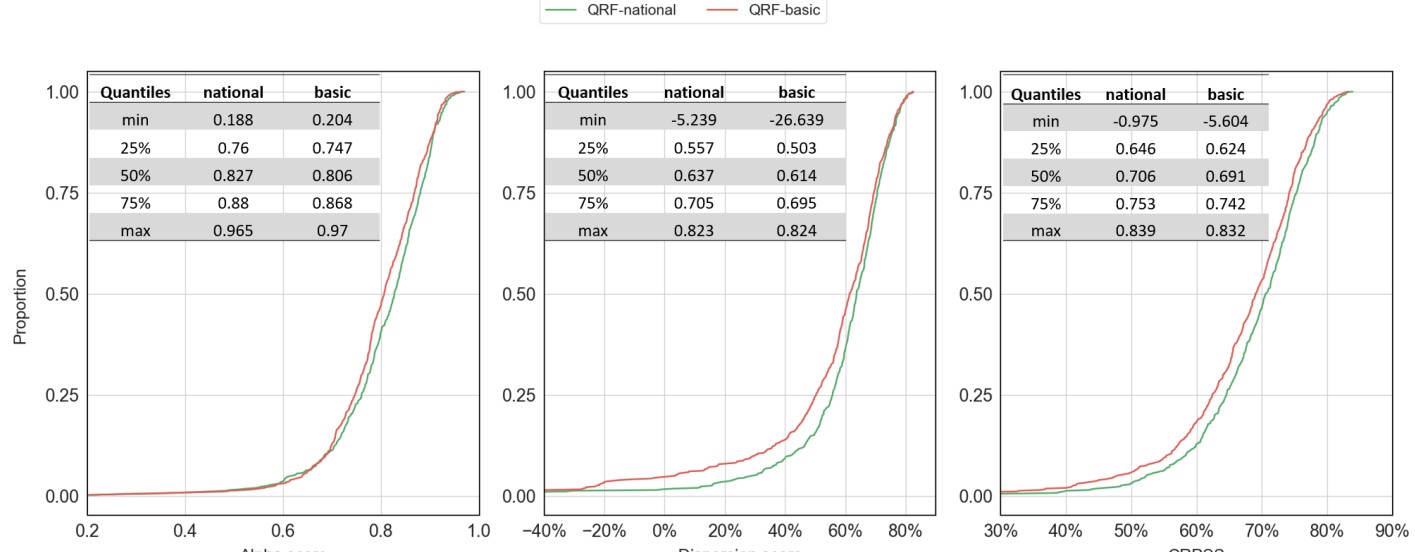

**Figure 8.** Distributional metrics across the 564 catchments for the QRF variants in the study. The red line represents the performance of QRF-basic and green represents QRF-national.

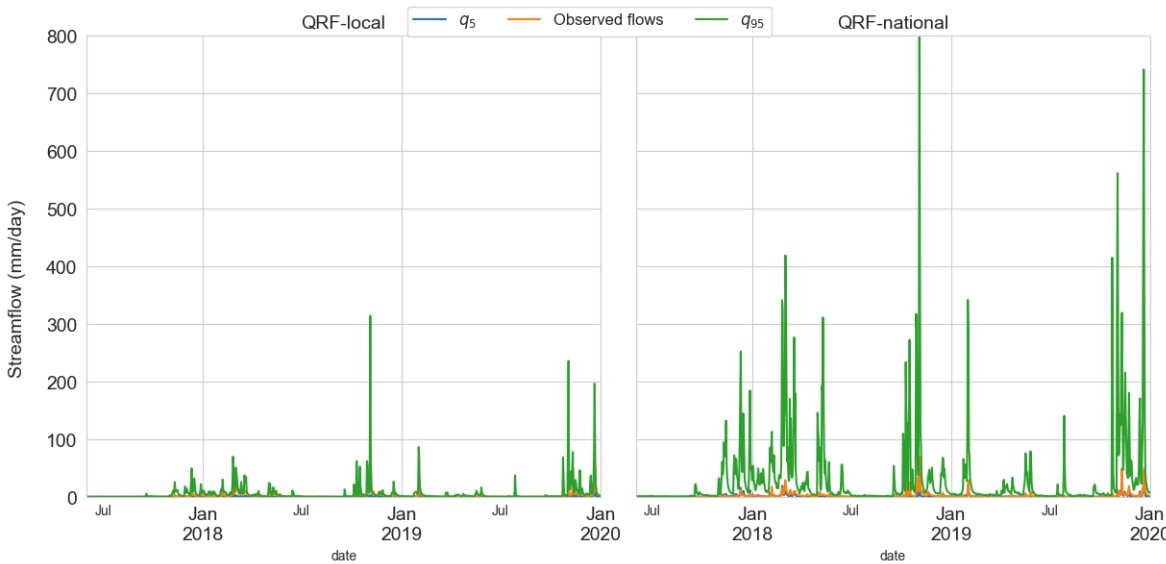

**Figure 9.** Example uncertainty estimates for a Mediterranean catchment (station Y960000102) from QRF-local (left) and QRF-national (right), covering the period from July 2017 to January 2020. The estimated $5\%$ quantile is shown in blue, and the $95\%$ quantile in green. The QRF-national model noticeably overestimates the upper ($95\%$) uncertainty quantile.





has important error variability but to a lesser degree (0.77< and <1.5), and group 3 (<0.77) which can be seen as having

normal variability. Values 1.5 and 0.77 are the 99% and 90% quantiles of the interquartile range used to standardize the errors $\epsilon_t$. QRF-national performed poorly for catchments with significant variations in the target variable, with notable decreases in reliability and CRPSS compared to a single-basin approach. These results highlight that robust standardization of input variables and errors is key to delivering meaningful multi-site QRFs, since this enables the algorithm to find analogs from locally calibrated hydrological model inputs. Furthermore, the aforementioned scale discrepancies occurred specifically for

catchments characterized by frequent zero values in simulated and observed streamflows. This can be problematic when using logarithmic relative hydrological errors. Figure. 9 illustrates QRF-local and QRF-national predictions for the 5% and 95% quantiles alongside observed flows for the Y960000102 catchment. Clearly, QRF-national overestimates the upper quantile. The local approach thus yields better results, since the error dynamics of this catchment are unconventional compared to the other catchments of the study.

**Table 4.** Average and median (shown in parentheses) differences between QRF-local and QRF-national models across distributional metrics, evaluated across different error scale groups. For catchments with extreme error variability (Group 1), QRF-national model degrades the quality of uncertainty estimates.

| Error scale | Alpha score | Dispersion score | CRPSS |
| :---: | :---: | :---: | :---: |
| Group 1 | -0.26 (-0.241) | -0.786 (-0.028) | -0.297 (-0.084) |
| Group 2 | -0.023 (-0.032) | -0.091 (-0.021) | 0.009 (0.011) |
| Group 3 | 0.015 (0.007) | 0.01 (0.008) | 0.021 (0.022) |

## 5 Discussion

### 5.1 When and where is it preferable to use a multi-site learning setup?

Although training QRF on local data yields good uncertainty estimates, as discussed in previous studies (Taillardat et al., 2016; Zhang et al., 2023), using a multi-site setup can slightly improve QRF performance. Our results indicate that the best improvements are achieved with QRF-national, which includes all 564 catchments of the study. The improvements mainly concern (i) catchments where local information is not sufficient (i.e., poor QRF-local performance), (ii) CRPSS and WSS scores for nearly all catchments, and (iii) periods of higher flows.

The results presented in Section 4 indicate that multi-site learning improves the performance of QRF models, and that larger models yield better uncertainty estimates (QRF-national > QRF-region > QRF-local). We find this result interesting, since one might argue that regional models in the QRF-region approach provide an equilibrium in spatial variability by aggregating similar catchments and, possibly, similar error dynamics (Johnson et al., 2023). Such an approach confines the QRF analog search to specific hydrological regions, without extending the search to the entire study area. QRF-national, however, is not constrained by the predefined hydro-climatological regions. Region indicators are included as input features for QRF-national,




and the algorithm is trained to find analogous events using these indicators, but is not strictly limited by them. Spatial variability appears to be beneficial for QRF, provided that appropriate catchment descriptors are used, and incorporating explicit measures
of catchment similarity could further improve multi-site learning with QRF (Hashemi et al., 2022; Kratzert et al., 2024).

Most improvements are noted for high and medium flows. QRF-national provided a better alpha score, i.e., reliability for 60% of the catchments during high and medium flows compared to QRF-local, but performance was identical for low flows. As highlighted in Bertola et al. (2023) and Auer et al. (2024), high flows are generally more difficult to predict and some high-flow events cannot be predicted exclusively from local historical data. QRF makes use of information from neighboring
catchments to provide uncertainty estimates for these events that can be more challenging to predict. Local information seems to be sufficient to characterize low-flow events.

### 5.1.1 What is the importance of meaningful catchment attributes?

We showed in Figure 8 that the improvements in QRF-national depend on the use of static descriptors. A nation-wide QRF variant with no catchment attributes (identical input descriptors to the local variant) performed worse than a classic single-
catchment QRF. This indicates that increasing hydrological diversity and lumping more catchment data are not the primary drivers of performance improvements. The information shared within a multi-site setup is best used by the QRF algorithm in conjunction with quality catchment descriptors. This would enable better uncertainty characterization and improved analog searches in similar catchments. The catchment descriptors used are readily available in the CAMELS-FR and other CAMELS datasets, making the use of such descriptors straightforward. Furthermore, a globally parametrized QRF post-processor is
able to extend its uncertainty estimates into ungauged catchments. Magni et al. (2023) found that RF is able to learn global mappings and improve deterministic estimates in poorly gauged and ungauged basins. Similar improvements could be obtained in an uncertainty estimation at ungauged catchments context, if appropriate catchment descriptors that do not rely on observed streamflows are selected.

### 5.2 On model complexity and computational time

Table 5 presents the number of parameters for each QRF variant, calculated as the product of the number of trees and the parameters of each tree (e.g., split thresholds, used input features). QRF-region and QRF-local exhibit a similar number of parameters, despite QRF-region providing better uncertainty estimates. In contrast, QRF-national shows a 47% increase in model complexity. This suggests that the performance gains of QRF-national come at the expense of increased computational cost which can be a drawback, especially since QRF stores not only the tree parameters but also samples used for training.
RF-based algorithms are CPU-intensive and suffer from memory voracity, especially for larger datasets Taillardat and Mestre (2020). In the case of our study, we had no difficulty fitting QRF-local and QRF-region with an Intel(R) Core(TM) i7-4770 CPU (3.40 GHz) and 16 GBs of memory. However, because of memory issues, we trained QRF-national on Jean-Zay HPC, where a single node with two CPUs (at 2.5 GHz) and 128 GBs of memory was sufficient. While training time is longer for multi-site settings, inference/prediction times are very similar to those of QRF-local.





**Table 5.** Cumulative number of parameters across all models.

| Model | Number of parameters |
|---|---|
| QRF-local | 379M |
| QRF-region | 364M |
| QRF-national | 551M |

**6 Conclusions**

In this study, we investigated the added value of multi-site learning with a hydrologically informed quantile random forest (QRF) post-processor across a large set of 564 French catchments. Three training setups were proposed – local, regional, and national – which we evaluated with different probabilistic metrics and across various hydrometeorological conditions. Based on reliability, sharpness, and overall metrics, our results indicate that multi-site learning improves QRF uncertainty
estimates, with notable enhancements; (i) for overall metrics (CRPSS and WSS) and deterministic metrics (NSE and KGE) (ii) at stations where the local approach provided unreliable uncertainty estimates; and (iii) for high and medium flows, where predictions can be more challenging. These findings corroborate previous studies (Fang et al., 2024; Bertola et al., 2023; Auer et al., 2024) that found that high-flow events can have similar characteristics in neighbouring catchments. These results suggest that the QRF algorithm in its regional extensions can leverage data from neighbouring catchments to improve its
uncertainty estimates; this is particularly advantageous given the off-the-shelf use of available catchment descriptors and the similarity of the learning process between local and regional variants. Additionally, the selection of representative and quality catchment attributes and static features is necessary to achieve the aforementioned improvements. We also found that using a single QRF post-processor for all catchments in the study (QRF-national) provided the best probabilistic predictions, which might indicate that the larger the model the better the uncertainty estimates with QRF. But QRF-national can yield erroneous
uncertainty estimates for catchments with significant scale variations in the errors. We argue that this is mainly due to the use of logarithmic transformation of relative errors, which strongly influences hydrological error dynamics at such stations. The use of other transformations and experimenting with other catchments groupings (Hashemi et al., 2022) could solve this issue. In addition, larger models are associated with higher computational costs, with increased complexity, and with a larger number of parameters. This is particularly relevant for QRF, as RF-based algorithms are known for their intensive memory use. However,
some solutions include the use of GPU-accelerated QRFs (Raschka et al., 2020).

We acknowledge certain limitations related to model-dependent artifacts. In this study, we were able to test QRF variants only using the GR6J hydrological model, as it was the only model for which simulations were available. However, the proposed framework is flexible and can be extended to other hydrological models and states. These findings highlight the performance enhancements of regional hydrologically informed QRF post-processing, and we aim to explore further in future studies the
merits of the proposed QRF framework in both forecasting applications and prediction at ungauged basins.





*Author contributions.* T.E. carried out the experiments and wrote the manuscript with support from F.B.; C.P. and V.A. helped review the manuscript and supervise the project.

*Code and data availability.* The quantile-forest package is available at https://pypi.org/project/quantile-forest. The airGR package can be downloaded from CRAN repositories using the following identifier: 10.32614/CRAN.package.airGR. The evalhyd-python package can be downloaded from the HAL open archive using the following identifier: hal-04088473. The CAMELS-FR dataset can be downloaded from the Recherche Data Gouv repository using the following identifier: 10.57745/WH7FJR.





**Appendix A: Hyperparameter tuning and libraries used**

Since we use QRF for probabilistic predictions, hyperparameter selection was based on the mean of the alpha score and CRPSS values. This would enable a selection based on the quality of overall uncertainty estimates with an emphasis on reliability. For
QRF-local, hyperparameters were tuned independently for each catchment and the set maximizing the aforementioned criteria during the validation period was selected. QRF-region and QRF-national hyperparameters were selected based on median criteria among the region's catchments. Overall, the selected hyperparameters were found to vary between catchments and regions. Table A1 presents the hyperparameters selected for optimization.

**Table A1.** Hyperparameters set optimized for QRF

| Hyperparameter | Values |
|---|---|
| Min samples leaf | 5, 10, 25, 50, 75, 100, 150, 200, 400, 600 |
| Number of estimators | 200, 400, 600 |
| Max features | sqrt, 8, 16 |
| Seeds | 0, 1, 2 |

**Appendix B: K-Nearest Neighbors model as benchmark**

**B1 K-Nearest Neighbours algorithm**

We used a naive k-nearest neighbors approach as a benchmark. The k-nearest neighbors (K-NN) algorithm is a non-parametric method that makes predictions based on the closest historical examples in the feature space. In hydrology, it is often used to estimate streamflow by averaging the outputs of the k most analogous conditions. Here analogs were used to estimate uncertainty, on a local basis. Table B1 presents the hyperparameters used for the K-NN algorithm, while table B2 compares the
approaches K-NN and QRF-local. Average and median (between parentheses) across the catchments of the study, including distributional, interval and deterministic metrics.

**B2 K-Nearest Neighbours and QRF-local comparison**

**Table B1.** Hyperparameters Set Optimized for K-NN approach

| Hyperparameter | Values |
|---|---|
| Number of neighbors | 5, 10, 25, 50, 75, 100, 150, 200, 400, 800 |
| Distance | Uniform, distance |

Table B2 compares the average and median performance metrics of QRF-local and K-NN across the 564 catchments. Overall, QRF-local consistently outperforms K-NN across all evaluated metrics. It achieves higher alpha scores, improved dispersion





scores, and better CRPSS values, indicating both more reliable and more skillful probabilistic predictions. These results suggest that QRF-local leverages dynamic input features more effectively than the simpler K-NN approach.

**Table B2.** Average performance scores across the 564 catchments for QRF-local and K-NN. Median scores are shown in parentheses. Bold values indicate better performance for each metric.

| Model | Alpha score | Dispersion score | CRPSS | CR90.0 | AWSS | WSS | NSE | KGE |
|---|---|---|---|---|---|---|---|---|
| K-NN | 0.771 (0.798) | 0.587 (0.63) | 0.655 (0.674) | 0.835 (0.848) | 0.678 (0.702) | 0.698 (0.72) | 0.825 (0.85) | 0.812 (0.835) |
| QRF_local | **0.799 (0.824)** | **0.593 (0.636)** | **0.674 (0.686)** | **0.860 (0.871)** | **0.680 (0.709)** | **0.715 (0.731)** | **0.832 (0.856)** | **0.822 (0.842)** |

## Appendix C: Assessment criteria

### C1    Continuous ranked probability score

Given a univariate predictive distribution $F$ and a corresponding realization $y$, the continuous ranked probability score (CRPS)
is defined as:

$$\text{CRPS}(F, y) = \int\limits_{-\infty}^{+\infty} (F(u) - H(u - y))^2 \, \mathrm{d}u, \tag{C1}$$

where $H$ is the Heaviside function such that $H(u-y) = 1$ if $u \geq y$ and 0 otherwise. In this study, the probabilistic predictions are in the form of draws distributions; hence, equation C1 has to be discretized for computation. We apply the method which is implemented in the function "CRPS_FROM_ECDF" from the Python package EvalHyd (Hallouin et al., 2023). The CRPS
is negatively oriented, meaning that smaller values are better.

### C2    Skill score

The performance of predictions can be more easily compared with that of a reference prediction skill scores. Skill scores (SS) are used to assess the relative quality of two predictions. They are generally defined as: $SS$ it is generally defined as:

$$\text{SS} = 1 - \frac{\overline{\text{S}}}{\overline{\text{S}_{\text{ref}}}} \tag{C2}$$

where $\overline{\text{S}}$ and $\overline{\text{S}_{\text{ref}}}$ are the scores of predictions from the model to evaluate and the reference model respectively. Climatology is commonly used as a reference. In this study, we consider climatological distributions of observed streamflows. It has been estimated as the empirical distribution of discharges across the training periods (P1).



### C3    Alpha score

Given a univariate forecast distribution $F_t$ and a corresponding realization $y_t$, the p value is $F_t(x_t) = p(Y_t \leq y_t)$ and the alpha
score is defined as:

$$\alpha'_y = \frac{1}{N_y} \sum_{i=1}^{N_y} \left| p_{y(i)} - p_{y(i)}^{(th)} \right|$$

where $p_y^{(i)}$ and $p_y(i)^{(th)}$ are the $i$th observed and theoretical p values of $y_t$ values. $N_y$ is the number of $y_t$ values. The alpha
score takes values between 0 and 1. It is positively oriented, with scores close to 1 reflecting perfect calibration.

The performance of streamflow forecasts can vary depending on the flow range considered (e.g., flood forecasting vs. drought
forecasting). Bellier et al. (2017) suggest a forecast-based sample stratification for continuous scalar variables in order to
consider the merits of streamflow forecasts on different ranges of flows. To ensure robust reliability estimates and prevent
potential compensation effects, the alpha score was calculated separately for three distinct flow ranges: low, high, and average
forecasted flows.

### C4    Dispersion score

Sharpness is quantified with the skill score of the forecast CRPS of median forecasts relative to climatological streamflow
distribution, in which CRPS median is defined as follows:

$$CRPS_{median}(F) = CRPS(F, F_{median}) \tag{C3}$$

where $F_{median}$ is the median value of the distribution $F$.

### Appendix D:  Results supplement





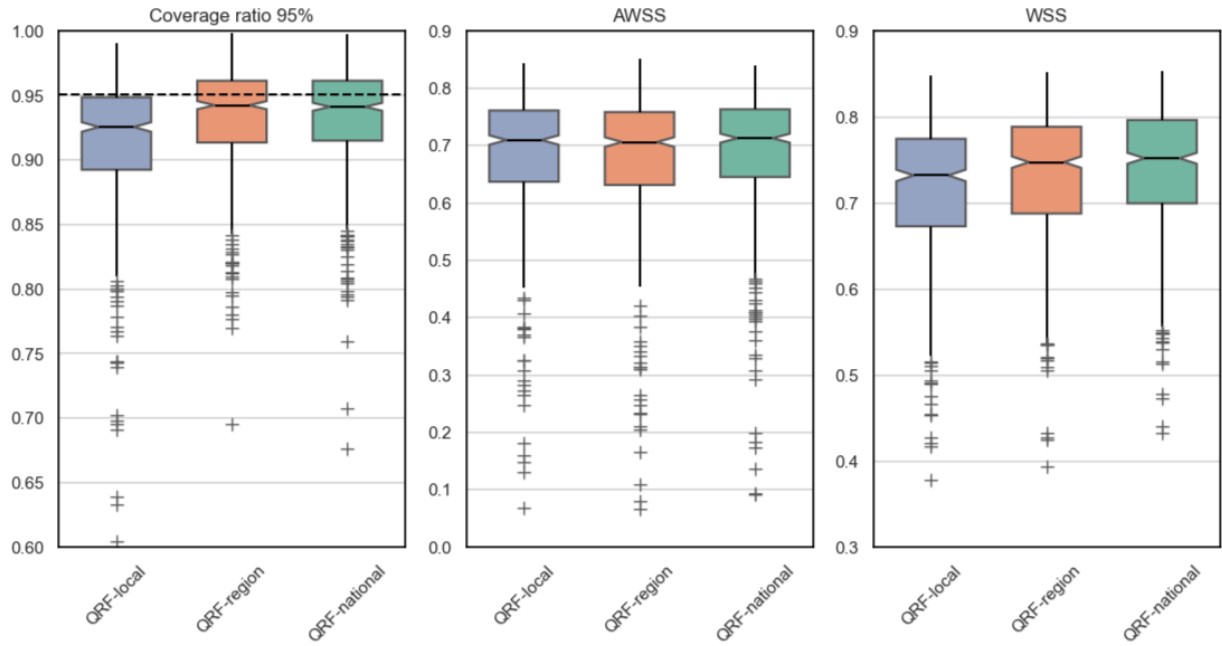

**Figure D1.** Box plots of 95% interval-based metrics across the 564 catchments of the study. Blue for the performance of QRF-local, orange for QRF-region, and green for QRF-national.



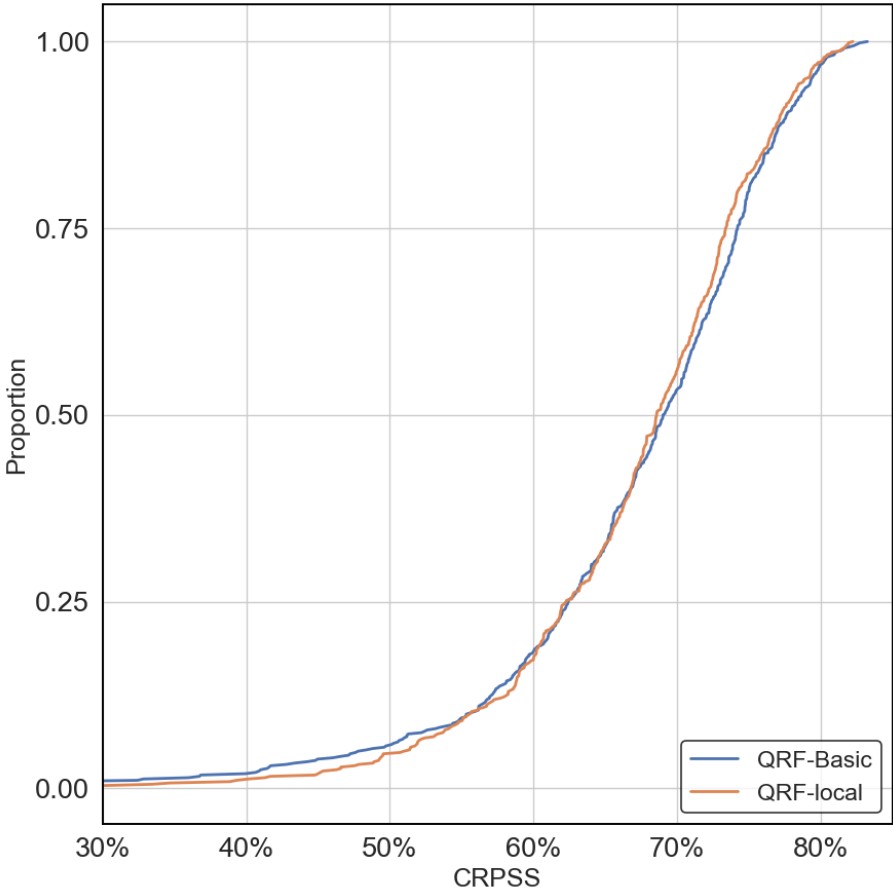

**Figure D2.** CRPSS metric across the 564 catchments. The blue line represents the performance of QRF-basic, orange represents QRF-local.

*Competing interests.* The authors declare that they have no conflict of interest.

*Acknowledgements.* We gratefully acknowledge Météo-France for providing the weather data and SCHAPI for the streamflow data. We would also like to thank the PREMHYCE and CIPRHES projects (ANR-20-CE04-0009), OFB, INRAE and SCHAPI for their financial support to the first author, which made this research possible. This work was performed using HPC resources from GENCI-IDRIS (Grant 2024-AD011013991R2).



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
