# Peer review of "Multi-site learning for hydrological uncertainty prediction: the case of quantile random forests"

_EGUsphere, 2025_

## Author Comment (AC2)

**Multi-site learning for hydrological uncertainty prediction: the case of quantile random forests**

Taha-Abderrahman El Ouahabi, François Bourgin, Charles Perrin, and Vazken Andréassian

https://doi.org/10.5194/egusphere-2025-3586

**Reply to Anonymous Referee #1**

Thank you very much for your comments on our manuscript, which will help improve our work. Please find below our replies (in blue) to your comments (in black), as well as how we intend to modify the paper to account for your suggestions and recommendations.

**Overall comments**

This is a consistently interesting and well-considered study on the benefits of using multiple sites to train a probabilistic machine learning method (QRF) to predict hydrological model errors. The study shows that the inclusion of multiple sites indeed does improve the performance of QRF. The study has clear aims, and its conclusions are well supported by rigorous cross-validation and forecast verification. This is a minor point, but I was quite taken with their innovative way of measuring sharpness using the CRPS, which neatly sidesteps the issue of having to focus on one or two intervals with average width of prediction intervals, the conventional method for assessing of sharpness (which can give contradictory results for different intervals, and of course tells you nothing about intervals that are omitted).

The finding that the use of multiple sites helps the QRF may in some ways seem obvious in retrospect, but this is the thing with the best studies: the findings often look obvious after the authors have presented them! Accordingly, I think the study makes a significant contribution to the literature on hydrological error modelling. Their further investigation of the use of regional/national methods of including sites provides practical guidance to anyone wishing to implement their methods, of which I am one.

For my own interests I would have liked to have seen a comparison with a more conventional error modelling technique - e.g. a simple AR1 model assuming Gaussian errors after transformation - but I understand that this would have considerably lengthened the study, and is not strictly within the aims of what the authors set out to do. So I am happy for this to be omitted. I have a few questions about methods in the specific comments, the most notable of which is whether static climatic/hydrologic predictors are cross-validated. Assuming the answer is 'yes', I recommend this study be published essentially in its present form, subject to technical corrections.

Thank you for the positive feedback on our work. We agree that a comparison with other and simpler methods would be interesting, but as mentioned we feel it is beyond the scope of this study. We will add a call for more comparative studies in our conclusion. Regarding the cross-validation of static predictors, we have provided some arguments in our response

below, but we remain open to perform additional experiments. This will involve re-computing the climatic predictors from the CAMELS-FR database and checking the impacts on our results.

Specific comments

L53 "They found that larger LSTM models trained on all available basins outperform smaller models trained on a limited set of catchments. This is because, for some ML approaches, models calibrated on larger training datasets can outperform smaller and more specialized models" This is effectively saying "LSTMs perform better on larger datasets because LSTMs perform better on larger datasets". Please avoid instances of circular reasoning like this. An additional point is that as far as I understand it LSTMs significantly outperform conventional rainfall runoff models for predictions in ungauged basins. This differs from applications where models are calibrated and used on the same catchment, where conventional rainfall-runoff models can perform similarly well to LSTMs. This may be worth mentioning.

Thank you for this feedback, we will modify the manuscript to improve clarity. We agree that LSTMs outperform conventional rainfall-runoff models for predictions in ungauged basins, but that the current situation is more balanced for predictions in gauged basins. This will be highlighted in the modified manuscript.

L83 "Potential evaporation (PET) is calculated using the formula proposed by Oudin et al. (2005)." Could the authors briefly list the inputs used in this formula?

Yes, PET formula is given by the following equation:

PET = 0.408 × Ra × (T + 5) / λ

If T ≤ 5°C, then PET = 0

Where:

- PET = potential evapotranspiration (mm/day)
- Ra = extraterrestrial radiation (MJ/m²/day)
- T = mean daily air temperature (°C)
- λ = latent heat of vaporization (≈ 2.45 MJ kg$^{-1}$)

Ra depends on the localization of the basin and Julian day values and the temperature is the only dynamical meteorological input used to estimate PET. We will include this in the revised version of the manuscript.

L84 "Since our interest is in developing a multi-site QRF post-processor, we used several static basin-averaged attributes describing climate, topography and geology." Would be good to foreshadow that these are listed in Table 1.

Thank you for this suggestion, we will add this information.

L108 "with a power of 0.5 and -0.5 prior transformations on streamflow" It's not clear to me where the power is being applied and what this transformation is. Please specify - in an appendix is fine.

The two power transformations are applied separately to streamflow values – both observed and simulated values – before calculating the two associated KGE criteria. We will add a reference to the recent study of [1], where the use of streamflow transformations for model calibration is investigated.

$$KGE(Q^{obs}, Q^{prd}) = 1 - \sqrt{(r(Q^{obs}, Q^{prd}) - 1)^2 + (\alpha(Q^{obs}, Q^{prd}) - 1)^2 + (\beta(Q^{obs}, Q^{prd}) - 1)^2}$$

The power transformation is applied on $Q^{obs}$ and $Q^{prd}$. This will be added to the appendix.

L154 Table 1. As is common, many of the static descriptors are based on climatic/hydrologic predictions (mean precip, PE, temp, etc.). I just wanted to confirm that these were cross-validated for this study: i.e., that they were computed separately for each of the training, validation and testing periods. As I'm sure the authors are aware, rigorously cross-validating predictors is an important aspect of testing any prediction system. I realise this kind of cross-validation is sometimes not done in ML studies, but it should be.

We understand the concerns and we agree that rigorously cross-validating predictors is important. As rightly pointed out, this is indeed not done in most of the ML studies we are aware of (e.g. [2]), and we followed this debatable common practice. We can understand both arguments: the need to cross-validating predictors and, on the other hand, the use of long-term climatic values taken as "static" descriptors. Our understanding of the role of the static descriptors in multi-site training is that they allow for creating some similarity-based relationship between catchments and that this relationship should be "static". We understand that some catchments have a non-stationarity behavior, but we feel that looking at the implications of non-stationarity is beyond the scope of this study.

L225 "The sharpness metric is the continuous ranked probability score (CRPS)" This is a really nifty way of measuring sharpness!

Thank you for your comment. It is quite unusual indeed, but we found in our previous works that it was an interesting alternative to other ways of measuring sharpness. It is related to an interesting decomposition of the CRPS proposed in the PhD thesis of [3] (in French). We will add the reference to the work of Bontron.

L258 "In terms of sharpness, the different QRF variants performed similarly, which is interesting given that multi-site setups significantly improve CRPSS values". Might be worth saying why it's interesting: it's quite possible (even unsurprising) that sharpness (a property of the prediction only) is the same but CRPSS (which considers the joint distribution of obs and predictions) differs.

Thank you for pointing this out: we intended to say that this shows that regional QRFs give more importance to reliability (as they consider the joint distribution of observations and predictions).

L272 Figure 4. Might be worth stating that curves that track closer to the right of the plots indicate better performance in the caption.

We will clarify this in the revised manuscript.

L286 "Table 3 summarizes the average values of the alpha, dispersion, CRPSS, and interval scores for three flow groups: high (> 67%Qsim), medium

(> 34%Qsim and < 66%Qsim), and low flows (< 33%Qsim)." Please confirm that performance scores are stratified based on when predictions exceed these thresholds, not when observations exceed them.

Yes, performances were stratified based on the median values of the probabilistic predictions. This will be clearly highlighted in the revised version of the manuscript.

L314 "Furthermore, the aforementioned scale discrepancies occurred specifically for catchments characterized by frequent zero values in simulated and observed streamflows." I wondered about this. Normalising errors with a log transformation is one thing, but maintaining normality in the presence of zeros in observations - and potentially also in the the simulations after the QRF is applied - is quite another. While this isn't a total solution, might it be helpful to consider the proportion of zero flows as a static predictor?

Yes, indeed. We will further discuss this in the following paragraph.

L320 Discussion. I would have liked to see a paragraph or two added that briefly discusses the following topics. However, I understand that my interests are not necessarily the authors' and also may not be interesting to a more general audience, so I leave it to the authors to decide which of these issues (if any) they may wish to discuss:

1. The weaknesses of the method (discussed throughout the manuscript) - e.g. application to ephemeral catchments - and how the authors might improve these.

2. The sensitivity of the method to data availability. The authors used the astonishingly comprehensive CAMELS-FR dataset, but many of us work in regions with only a fraction of this gauge coverage. e.g. What would have happened if they only had access to 50 gauges in their dataset? What might have happened if observations are concentrated on a particular hydrological type, but applied outside this type?

3. Are there prospects for applying this method to produce reliable probabilistic predictions in ungauged basins?

We plan to include the following paragraphs in the revised manuscript (section 4.6), in order to discuss these issues:

The provided analyses highlighted some limitations of the multi-site QRFs. The first concerns ephemeral catchments which are characterized by zero flow values. Modelling

ephemeral catchment dynamics is generally challenging [4, 5], and this is especially the case in our study with the use of the log-based error transformation. Figure 9 of the results section clearly showed that multi-site learning can degrade the predictions for ephemeral catchments as they often overestimate uncertainty. Although we have added the $\delta$ offset parameter, the use of an alternative transformation that is less sensitive to zero flow values (e.g., a Box–Cox transformation) could better stabilize hydrological errors used to train the QRF models. Another pragmatic solution could be the treatment of such catchments separately when training multi-site QRF. As showed in Figure. 9, QRF-local better managed the case of zero flow values. In the literature, other approaches [6, 7] use adapted catchment groupings based on other attributes (climatic, etc.) or a statistical clustering approach classifying homogenous catchments together. Similarly, the use of the number of zero flow values as input feature could also help QRF to better distinguish catchments characterized by this issue, and help QRF to find adequate analogous events.

We used a large sample dataset (CAMELS-FR) for models training, but many practical hydrological applications only have access to a limited number of gauges for training purposes. The generalizability of uncertainty estimates of QRF to catchments outside its training region was treated in various studies. We did not explicitly test this, but we believe that the generalizability of regional QRF variants depends on the similarity between hydrological errors for catchments where the training occurred and regions to which extrapolation of the uncertainty estimates will be carried out. Finally, the proposed framework can be adapted for a prediction at ungauged basins, and a proper spatio-temporal cross-validation experiment [7] would be needed to verify this. The main practical difficulty lies in obtaining consistent hydrological model states for ungauged catchments, and to adapt the significant additional uncertainty usually associated with such settings [8, 9, 10]. If this can be properly handled, the proposed QRF multi-site variants could provide meaningful uncertainty estimates for the context of uncertainty estimation for ungauged catchments.

L362 "However, because of memory issues, we trained QRF-national on Jean-Zay HPC, where a single node with two CPUs (at 2.5 GHz) and 128 GBs of memory was sufficient." I'd be interested to know how long the parameters estimation took on this hardware. Cloud computing means that many now have access to large computers, but the run-time can still make these resources expensive.

For each parameter, it takes around 25 minutes for QRF-national.

L141 "But, it is important to note that these scale features are not available" suggest deleting 'But,'

Grammar and terminology will be checked in the final version of the manuscript. Thank you very much for your comments.

*Bibliography*

[1] Thirel, Guillaume, et al. "On the use of streamflow transformations for hydrological model calibration." Hydrology and Earth System Sciences 28.21 (2024): 4837-4860. https://doi.org/10.5194/hess-28-4837-2024

[2] Auer, A., Gauch, M., Kratzert, F., Nearing, G., Hochreiter, S., and Klotz, D.: A data-centric perspective on the information needed for hydrological uncertainty predictions, Hydrol. Earth Syst. Sci., 28, 4099–4126, https://doi.org/10.5194/hess-28-4099-2024, 2024.

[3] Bontron, G., 2004. Prévision quantitative des précipitations: Adaptation probabiliste par recherche d'analogues. Utilisation des Réanalyses NCEP/NCAR et application aux précipitations du Sud-Est de la France (Doctoral dissertation, Institut National Polytechnique Grenoble (INPG)). https://tel.archives-ouvertes.fr/tel-01090969

[4] McInerney, D., Kavetski, D., Thyer, M., Lerat, J. and Kuczera, G., 2019. Benefits of explicit treatment of zero flows in probabilistic hydrological modeling of ephemeral catchments. Water Resources Research, 55(12), pp.11035–11060. https://doi.org/10.1029/2018WR024148

[5] Li, M., Wang, Q.J., Bennett, J.C. and Robertson, D.E., 2016. Error reduction and representation in stages (ERRIS) in hydrological modelling for ensemble streamflow forecasting. Hydrology and Earth System Sciences, 20(9), pp.3561–3579. https://doi.org/10.5194/hess-20-3561-2016

[6] Hashemi, R., Brigode, P., Garambois, P.A. and Javelle, P., 2022. How can we benefit from regime information to make more effective use of long short-term memory (LSTM) runoff models? Hydrology and Earth System Sciences, 26(22), pp.5793–5816. https://doi.org/10.5194/hess-26-5793-2022

[7] Fang, S., Johnson, J.M., Yeghiazarian, L. and Sankarasubramanian, A., 2024. Improved national-scale above-normal flow prediction for gauged and ungauged basins using a spatio-temporal hierarchical model. Water Resources Research, 60(1), e2023WR034557. https://doi.org/10.1029/2023WR034557

[8] Razavi, T. and Coulibaly, P., 2013. Streamflow prediction in ungauged basins: review of regionalization methods. Journal of Hydrologic Engineering, 18(8), pp.958–975. https://doi.org/10.1061/(ASCE)HE.1943-5584.0000690

[9] Oudin, L., Andréassian, V., Perrin, C., Michel, C. and Le Moine, N., 2008. Spatial proximity, physical similarity, regression and ungaged catchments: A comparison of regionalization approaches based on 913 French catchments. Water Resources Research, 44(3). https://doi.org/10.1029/2007WR006240

[10] Bourgin, F., Andréassian, V., Perrin, C. and Oudin, L., 2015. Transferring global uncertainty estimates from gauged to ungauged catchments. Hydrology and Earth System Sciences, 19(5), pp.2535–2546. https://doi.org/10.5194/hess-19-2535-2015

---

## Author Comment (AC3)

**Multi-site learning for hydrological uncertainty prediction: the case of quantile random forests**

Taha-Abderrahman El Ouahabi, François Bourgin, Charles Perrin, and Vazken Andréassian

https://doi.org/10.5194/egusphere-2025-3586

**Reply to Derek Karssenberg's comments**

Thank you very much for your comments on our manuscript, which will help improve our work. Please find below our replies (in blue) to your comments (in black), as well as how we intend to modify the paper to account for your suggestions and recommendations.

The manuscript proposes and evaluates the use of quantile random forests for correction of streamflow predicted with a process-based model. The main innovation compared to previous studies on streamflow error correction is the use of quantile random forests as it provides a means to estimate uncertainty in corrected streamflow. Also, unlike previous studies, this study extensively compares results for approaches that use local, regional, or national (France) data for training the error correction model. In my opinion this is a very interesting study. The methodology is state of the art, and the manuscript is relevant to development of error correction methodology (also in other domains).

My comments mainly refer to how the study is presented while I suggest a number of relatively small additions. Please find below my main comments followed by a list of minor comments.

**Introduction**

The introduction needs some revision to further increase the impact of the study and to make it more accessible. The problem definition needs to be defined more completely and more precisely. It remains unclear what the 'simulation context' (used in the methods section, line 125) is. In my opinion it is important to clearly state that this paper is about error correction of process-based model predictions, in the situation/context where predictions are made without relying on extrapolation of past observations of streamflow (for short range (small lead time) forecasts this would be more powerful). Also, the paper is not about prediction for ungauged catchments as all models are trained on historical streamflow at the location for which predictions are made. The simulation context thus is, I think, mainly reconstructing or projecting (e.g. under scenarios of climate change) streamflow for catchments that have streamflow available.

Also, the second contribution (line 62, spatial catchment descriptions) does not come with a clearly substantiated problem addressed by this contribution.

Please clearly describe what is meant by 'regional' in 'regional learning', 'regional approaches', 'regional bias', 'regional post-processing', etc. It is central to the study but it is not clearly defined. Is 'multi-site learning' (line 70) the same (please explain in manuscript).

Please clearly describe what is error corrected (i.e. streamflow from a process-based model). This is not clearly stated in the introduction (e.g. line 64 'model states', model states from what?).

What are 'spatially varying catchment characteristics? Line 64. Please explain or rephrase.

By 'simulation context', we mean the reconstruction/projection of streamflow for catchments that have streamflow available. We will clarify the manuscript in order to precisely introduce the problem definition. Used terminology and syntax will be introduced more clearly.

**Hyper parameter tuning – metrics used**

I suggest moving information from Appendix A (line 398 – 403) to the main text (Methods), in particular the fact that hyperparameter tuning is done on metrics that refer to probability distributions (instead of deterministic ones). To my knowledge this study is ***quite unique in doing so*** (but I may be wrong but even then I would still move it to the main text). It is also suggested to state in the main text that in the local modelling, hyperparameters are different between catchments (which should further improve the results for the local model compared to an approach fixing hyperparameters across catchments).

We have moved the section. We have also added the section that used to be in the Appendix to highlight that QRF-local hyperparameters are fitted for each catchment individually.

**Assessment criteria**

The assessment criteria are well chosen. However, I think it can be presented better in the Methods and Results section.

First, I suggest giving additional explanation on the terminology. If I am correct 'sharpness' refers to the magnitude of the uncertainty of the prediction, i.e. the lower the better. Please try to explain this more extensively as not all readers will be familiar with this term. The term 'reliability' in the context of your manuscript refers to whether the modelling approach is capable of providing correct estimates of the uncertainty (preferably the complete distribution should be correct).

Second, I suggest then to somewhat more clearly explain to what (sharpness or reliability) each metric refers. For instance, both the alpha score and coverage refer to 'reliability'. Connecting these metrics could also be done in the Results section; e.g. one would expect similar results (relative performance between local, regional, national) for alpha and coverage ratio as these both refer to reliability. This is not stated at all in the Results and the reader has to make it up by herself.

Third, please be precise in the explanation. For instance 'It calculates the closeness of predicted uncertainty distributions to the statistical distribution of observed streamflows' (line 221) reads like you compare the distribution of the error term (from the model) with distributions of streamflow (over time?). This is not at all the case! Instead alpha is a metric summarizing the QQ plot, which is really a probabilistic property (as the authors will certainly be aware of). Also, please use correct units (for instance, CRPSS is given as percentage in the figure while in the main text it is in the range of 0-1 it seems).

We agree the original wording was unclear. Indeed, sharpness refers to the magnitude of the uncertainty estimates, and normally, lower values of sharpness refer to better uncertainty estimates. The used metric however was transformed to a skill score, which made values of the Dispersion score closer to 1 better than lower score values (similarly to CRPSS). We have revised the text to better explain the definitions of reliability and sharpness, in addition to clearly highlighting the better performances direction for the used metrics. The link between the alpha score and coverage ratio will also be clearly highlighted.

**Results**

Please summarize the results of the hyper parameter tuning.

We conducted a hyperparameter grid search for each QRF variant and used the aforementioned probabilistic metric combining both Alpha score and CRPSS. These hyper-parameters include the size of the forest (number of trees), the minimum number of samples at child nodes, and the maximum number of candidate variables to use for splitting at each tree node. Each hyperparameter search was repeated with three seeds and the median score was calculated to obtain a more robust selection. Overall, the performances of QRF were most sensitive to the minimum number of samples at child nodes. Using the dataset described above, QRF was trained with minimum number of samples at child nodes ranging from 5 to 600 data points. The following figure shows the impact of hyperparameters on the selection metric for QRF-national:

[Figure]

[Figure]

***New figure 1****: Hyper-parameters optimization results for QRF-national. The selection criterion is the median hyperparameter criterion across the catchments of the study.*

It is notable that best results were recorded for lower values of the hyper-parameter 'minimum samples at leaf nodes'. The improvement slows for values lower than 25, and the mean scores were within one standard deviation of the mean scores for the other values. It is worth mentioning that very low values of minimum samples at leaf nodes might result in overly complex QRF models. As such, a minimum samples at leaf nodes of 10 is selected. Overall, QRF was found to be fairly insensitive to the number of candidate predictors used for splitting at each node. By default, the quantile-forest library uses the integer value of the square root (sqrt) of the total number of predictors for this parameter. With 31 total predictors for QRF-national, 6 would be the default and the previous figure showed that using the default value of square root was slightly better. For the number of trees parameter, a forest with more trees will generally be more skillful than one with fewer trees, as it can fit more on the nuances of the training set, and there is a point when the rate of improvement with more trees is negligible, as noted in [1, 2]. Most of the boxplots ranges overlap, and it seems that the results are not overly sensitive to this QRF parameter. For the experimented values, the above Figure shows that a number of 400 trees allows for slightly better performances.

We followed a more automatized approach for the selection of the hyperparameters of the other QRF-variants. And the distribution of the selected hyper parameters can be found in the appendix.

**Magnitude of uncertainty (sharpness)**

It seems the modelling approach underestimates uncertainty for all scenarios. This is an important outcome. Please add this information to the Results (it is not mentioned at all) and provide possible explanations in the Discussion section. One possible cause is the fact that the approach neglects uncertainty in the streamflow prediction of the process-based model.

It is true, there are certain catchments where uncertainty is severely underestimated. It is worth mentioning that negative values rather denote overestimation. This is linked to the

existence of 0 values for certain catchments which makes the estimation of uncertainties more difficult.

**Process-based model as benchmark**

I am aware the main question of the manuscript is not on how much error correction contributes to improved streamflow prediction compared to using streamflow from the process-based model (without error correction). However I am in the opinion that it is still extremely interesting to add information (and if possible a short discussion) on the performance of the process-based model before adding the error correction. This could be done by adding curves for this process-based only benchmark to figures, or values in tables, or values in the main text. This would also allow you to compare the results regarding the improvement of streamflow prediction after error correction with those in Magni (2023), Shen (2022) and possibly others. It the improvement in your study comparable to other studies?

We agree with this element and we propose to add the following paragraph to section 4.3 (Deterministic metrics). It compares mean QRF estimates with the original GR6J predictions:

[Figure]

***New Figure 2.*** *CDF of deterministic metrics across the 564 catchments for the QRF variants in the study during the testing period. The blue line represents the performance of QRF-local, orange represents QRF-region, green represents QRF-national, while red the performance of raw GR6J predictions.*

Figure 1 also provides deterministic metrics for the raw GR6J predictions. The figure highlights that the proposed QRF methods can improve hydrological deterministic predictions, especially for NSE. For example, QRF-national produced better NSE

performances compared to GR6J predictions (0.87 vs 0.86 in median NSE) and for 75% of the study's catchments. Overall, QRF variants had better NSE for the majority of the catchments. For KGE, the raw GR6J estimates outperforms all tested QRF approaches. We expected that the proposed variants would also improve KGE as was observed in the previous studies of [3, 4, 5]. For example, [4] used a closely related deterministic RF framework for hydrological error correction and found that the hybrid RF approach boosted streamflow predictions from a median of -0.03 to 0.51. Here, the post-processing was not beneficial for KGE performances. We argue that this can be in part attributed to how QRF hyper-parameters were selected. The used statistical criterion in this study aims to maximize the probabilistic performances of reliability and sharpness of the uncertainty estimates. While for the aforementioned studies, the RF post-processor was optimized specifically for the KGE criterion.

**Title**

Consider revising such that it also covers the fact that this manuscript is about error-correction (or combining process-based modelling and machine learning – sometimes referred to as hybrid modelling). I agree that the case is quantile random forests but the case is also error correction (maybe more so).

Thank you for highlighting this point. We propose to modify the title as follows:

Multi-site learning for hybrid error-correction: using quantile random forests for hydrological uncertainty prediction

**Minor comments**

Line 64 What does 'For this..' refer to?

Figure 1 Please add a scale bar.

These point will be addressed in the revised version.

Line 105 Please state what parameters were calibrated.

The following parameters of GR6J were calibrated:

- X1 – Production store capacity [mm]

   Controls the maximum soil moisture storage capacity of the production reservoir.

- X2 – Groundwater exchange coefficient [mm/day]

   Governs the rate of exchange between groundwater and the streamflow system.

- X3 – Routing store capacity [mm]

Represents the capacity of the non-linear routing reservoir responsible for routing.

- X4 – Time constant of unit hydrograph [days]

  Controls how quickly water is routed through the hydrograph

- X5 – Groundwater exchange threshold [–]

  Influences the sign of groundwater exchange.

- X6 – Exponential store coefficient [mm]

  Relates to the capacity/depletion of the exponential store, which improves low-flow simulation.

These parameters will be included in the revised version of the manuscript.

Line 105 'prior transformations' What is meant by this?

The two power transformations are applied separately to streamflow values – both observed and simulated values – before calculating the two associated KGE criteria. We will add a reference to the recent study of [6], where the use of streamflow transformations for model calibration is investigated.

$$KGE(Q^{obs}, Q^{prd}) = 1 - \sqrt{(r(Q^{obs}, Q^{prd}) - 1)^2 + (\alpha(Q^{obs}, Q^{prd}) - 1)^2 + (\beta(Q^{obs}, Q^{prd}) - 1)^2}$$

The power transformation is applied on $Q^{obs}$ and $Q^{prd}$. This will be added to the appendix.

Line 125

It is stated here that in the simulation context of this study, streamflow is not available. This is not really true. The manuscript describes a methodology that only applies to locations where streamflow is available (for training, validation). For testing (or projections/reconstruction) I agree it can be done without measured streamflow (for the timesteps for which testing is done) but in this manuscript, results/testing metrics are only presented for locations where streamflow was used for training (i.e. this is not an ungauged catchment study). This is in my opinion not an important limitation, but it has to be clearly stated what this study is about (please refer to my comments related to the introduction).

Indeed, we were not clear with this element. The context of the proposed manuscript does not target the prediction at ungauged catchments context -but this will be specifically treated in an additional paragraph in the discussion section. We will clear out any mis-understanding regarding this point and highlight that the proposed methodology necessitates observed flows (to calibrate the hydrological model and calculate errors) and is geared towards cases of projections and reconstruction.

Line 130, 'production'

What is meant here?

We intended to refer to the production store of the hydrological model. This point will be clarified in the revised version of the manuscript.

Line 131, 'moving averages'

What was the filter size?

Moving average filter size is equal to the catchment response time, which was obtained from the X4 parameter of the used hydrological model. This hydrological parameter is related to the timing of the catchment's response to rainfall. We will add this information in the revised version of the manuscript.

Line 194

Refer to Figure 1

Line 236

Number -> proportion

The remaining points will be addressed and corrected in the revised version. Thank you very much for your comments.

*Bibliography*

[1] Oshiro, Thais Mayumi, Pedro Santoro Perez, and José Augusto Baranauskas, 2012. How many trees in a random forest? In: International Workshop on Machine Learning and Data Mining in Pattern Recognition. Berlin, Heidelberg: Springer-Verlag. https://doi.org/10.1007/978-3-642-31537-4_13

[2] Breiman, Leo, 2001. Random forests. Machine Learning, 45(1), pp.5–32. https://doi.org/10.1023/A:1010933404324

[3] Zhang, Yuhang, Aizhong Ye, Bita Analui, Phu Nguyen, Soroosh Sorooshian, Kuolin Hsu, and Yuxuan Wang, 2023. Comparing quantile regression forest and mixture density long short-term memory models for probabilistic post-processing of satellite precipitation-driven streamflow simulations. Hydrology and Earth System Sciences, 27(24), pp.4529–4550. https://doi.org/10.5194/hess-27-4529-2023

[4] Shen, Y., Ruijsch, J., Lu, M., Sutanudjaja, E.H. and Karssenberg, D., 2022. Random forests-based error-correction of streamflow from a large-scale hydrological model: Using model state variables to estimate error terms. Computers & Geosciences, 159, p.105019. https://doi.org/10.1016/j.cageo.2021.105019

[5] Magni, M., Sutanudjaja, E.H., Shen, Y. and Karssenberg, D., 2023. Global streamflow modelling using process-informed machine learning. Journal of Hydroinformatics, 25(5), pp.1648–1666. https://doi.org/10.2166/hydro.2023.217

[6] Thirel, Guillaume, et al. "On the use of streamflow transformations for hydrological model calibration." Hydrology and Earth System Sciences 28.21 (2024): 4837-4860. https://doi.org/10.5194/hess-28-4837-2024